# Design strategies for tourism cultural and creative products of Hetuala city based on KANO and AHP

Zhaolong Liu*, Honghe Gao

School of Art and Design Liaoning Petrochemical University Fushun, Liaoning, China

* 84363335@qq.com

## Abstract

With the rapid development of China's "Culture + Tourism" industry, tourism cultural and creative products have increasingly become a key factor in attracting tourists. However, the tourism cultural and creative products in Hetuala City have difficulty effectively attracting tourists and meeting diverse consumer demands. To address this issue, this paper proposes a user-centered consumption demand evaluation framework, aimed at providing systematic guidance and decision-making support for the design and development of tourism cultural and creative products in Hetuala City. Firstly, consumption demands for tourism cultural and creative products were collected through interviews, initially identifying 80 demand indicators. Expert interviews were then employed to further summarize and refine these into 29 core demand indicators. Based on the 29 demand indicators, a sample survey of 150 consumers was conducted using the KANO model. Data analysis identified five must-be needs, nine performance needs, and ten attractive needs, thereby constructing a demand evaluation system for tourism cultural and creative products. Secondly, the Analytic Hierarchy Process (AHP) was applied to calculate the weights and importance rankings of each demand indicator, ensuring alignment between product design objectives and users' core demands. The results indicate that the evaluated demand attributes significantly influence tourist satisfaction, with regional characteristics, historical culture, and practical functionality regarded as key indicators, while tourists' expectations for product storytelling were particularly prominent. The integrated application of the KANO model and AHP method facilitates more accurate identification and evaluation of the importance of tourist demands, rendering the design of tourism cultural and creative products more systematic and precise. This study offers targeted design recommendations and optimization directions for the development of tourism cultural and creative products.

**Data availability statement:** All relevant data are within the manuscript.

**Funding:** This work was supported by Educational Commission of Liaoning Province of China (No. W2019013), Liaoning Provincial Education Science "13th Five-Year Plan" Period (No. JG20DB276).

**Competing interests:** The authors have declared that no competing interests exist.

# 1 Introduction

Cultural resources constitute the core of cultural industry development [1]. Regional culture is not only an essential component of national culture but also a driving force behind the comprehensive development of regional economies [2]. The development of each regional cultural industry is founded on the thorough exploration and utilization of local cultural resources [3]. As an important medium for presenting local culture externally, regional tourism cultural and creative products have gradually attracted growing attention from consumers [4]. Their significance is no longer confined to the functional attributes of the products themselves but also lies in enabling people to experience the distinctive connotations of local culture through these products [5]. Therefore, how to integrate regional characteristics into regional tourism cultural and creative products, and how to enhance the appeal of regional culture as such products become increasingly internationalized, has become a pressing issue to be addressed. This study is based on the historic ancient Hetuala City in China, exploring its traditional cultural heritage and transforming regional culture into creative products. It develops new designs that integrate traditional charm, ethnic characteristics, and cultural connotations, thereby creating value-added industries to promote regional economic and cultural development [6,7].

Hetuala City, located in Yongling Town, Xinbin Manchu Autonomous County, Liaoning Province, served as the capital established by Emperor Nurhaci, the founder of the Qing Dynasty. It is recognized as the birthplace of the Qing Dynasty and an important origin of Manchu culture, with a history spanning more than 400 years. The city preserves 33 ancient architectural sites, including the former imperial palace and Eight Banner military barracks, which embody profound historical heritage and significant ethnic cultural value. In 2002, it was designated as a National 4A-Level Tourist Attraction, and in 2006, it was listed as a National Key Cultural Heritage Protection Unit [8]. In recent years, driven by the "Culture-Driven Rural Revitalization" strategy, Hetuala City has leveraged its Manchu cultural resources and surrounding historical sites to develop a diversified cultural tourism system integrating sightseeing, folk customs, cuisine, and leisure. During the seven-day National Day holiday in 2024, the city received 205,000 visitors and generated revenue exceeding 70 million yuan. Its annual tourist volume surpassed one million, demonstrating strong potential for cultural tourism consumption.

Against the backdrop of the deepening integration of culture and tourism, tourism cultural and creative products have emerged as a vital medium linking local culture with market consumption [9]. Tourism cultural and creative products that center on regional culture while incorporating modern design concepts not only enhance the visitor experience but also play a crucial role in disseminating culture and shaping brand identity [10]. In the context of a continuously expanding tourism market, the development quality of Hetuala City's tourism cultural and creative products directly affects the site's cultural value output and its potential for economic growth. To gain a comprehensive and detailed understanding of the current state of Hetuala City's cultural and creative products, a market survey was conducted through questionnaires

and on-site visits to local tourism cultural and creative product shops. The findings reveal that the current market for tourism cultural and creative products exhibits the following characteristics:

First, the product variety is relatively limited. The tourism cultural and creative products available in the market are confined to a few categories such as stationery, bookmarks, and dolls, lacking deep integration with Hetuala City's rich cultural resources.

Second, existing products are overly simplistic in content expression and design form, lacking cultural connotation and distinctiveness. They fail to establish representative cultural symbols and cannot effectively convey Hetuala City's historical value and regional characteristics.

Third, most current products are low-cost industrial goods purchased through wholesale channels, lacking professional design teams or systematic support from the cultural and creative industry. This results in a lack of innovation and differentiation, weak regional characteristics, and serious homogenization, making it difficult to meet the growing demands of cultural tourism consumers.

In summary, the existing tourism cultural and creative products in Hetuala City exhibit significant shortcomings in product variety, content presentation, and integration of regional culture, making it difficult to meet the increasingly diverse consumer demands. These issues directly limit the effectiveness of tourism cultural and creative products in attracting tourists, enhancing cultural identity, and promoting tourism consumption, thereby affecting their cultural dissemination capacity and market competitiveness. Therefore, the core objective of this study is to enhance the market adaptability and cultural value of tourism cultural and creative products through innovative design approaches combined with precise user demand analysis. To achieve this goal, the study focuses on two key issues:

(1) How to scientifically and systematically identify and classify users' specific demands for tourism cultural and creative products;

(2) How to quantify the importance of different types of demands and effectively translate them into actionable design parameters to guide product development practices.

To this end, this study employs the KANO model to classify user demands and combines it with the Analytic Hierarchy Process (AHP) to weight and rank the importance of each demand type, thereby constructing a user-centered design framework that provides theoretical support and methodological guidance for the innovative development of tourism cultural and creative products in Hetuala City.

## 2 Literature review

### 2.1 Cultural resources of Hetuala city

Hetuala City is located in Xinbin Manchu Autonomous County, Liaoning Province. It was originally built in 1440 during the Ming Dynasty and expanded in 1603 by Nurhaci, the founding emperor of the Qing Dynasty, becoming the last mountain-style capital city in China.It is now designated as a National 4A Tourist Attraction and a Major Historical and Cultural Site Protected at the National Level. As one of the best-preserved Jurchen mountain cities, it serves as vital evidence for studying pre-Qing history and Manchu culture [11]. The scenic area comprises the "Hetuala Ancient City" and the "Chinese Manchu Folk Culture Park," preserving 33 historical sites and attractions, including the former imperial palace, Eight Banner military barracks, ancestral temples, and ancient market streets, systematically showcasing pre-Qing history and ethnic life. Its cultural resources encompass both tangible and intangible heritage, including Manchu cuisine, dwellings, clothing, religion, festivals, crafts, legends, and folk customs, reflecting distinct ethnic characteristics. Based on literature review, field investigation, and surveys, a tourism cultural resource map of Hetuala City has been developed (Fig 1). This rich and multidimensional cultural resource system provides a solid content foundation for the integration of local culture and tourism, the dissemination of ethnic culture, and the development of tourism cultural and creative products.

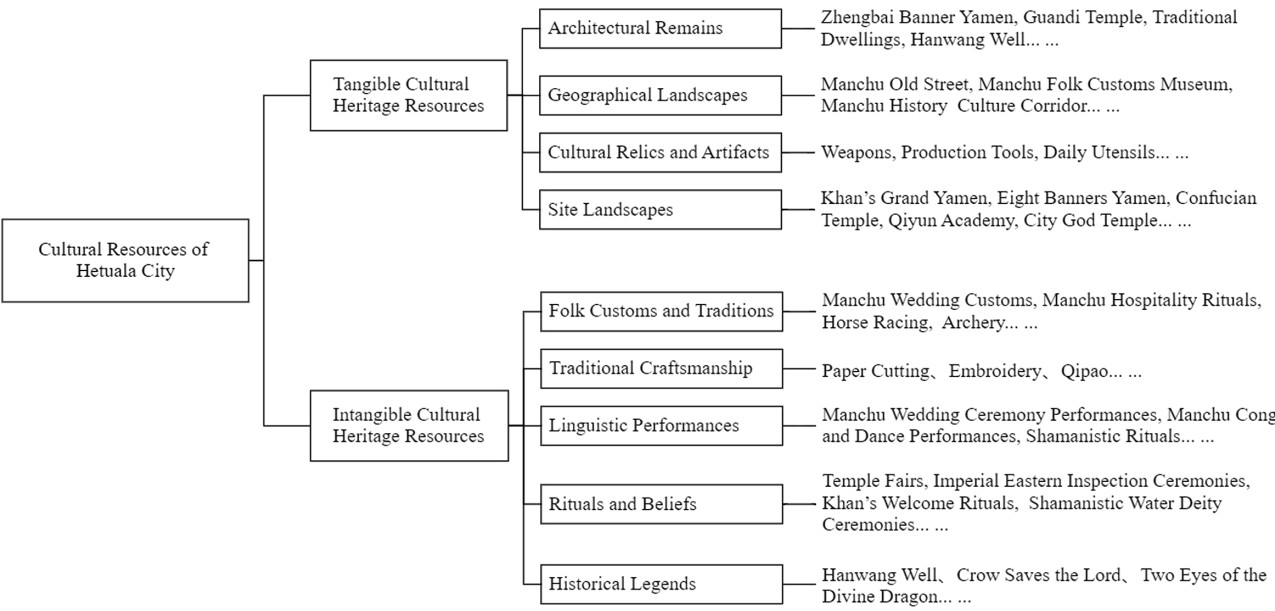

**Fig 1. Cultural resource map of Hetuala city.**

Researchers generally believe that Hetuala City holds great potential for transformation from a traditional cultural heritage site into a comprehensive ethnic cultural tourism destination. The diversity of its cultural resources, the narrative appeal of its historical stories, and the recognizability of its ethnic culture not only enhance the cultural value of the tourism experience but also provide fertile "cultural soil" for the thematic refinement, symbolic design, and market transformation of tourism cultural and creative products [1]. Sui Chao argues that Hetuala City not only possesses abundant Manchu tourism resources but also holds profound cultural depth, giving it unique advantages in the development of Manchu cultural tourism. The development of Manchu cultural tourism in Hetuala City will lay a solid foundation for the overall growth of the tourism industry in Liaoning Province [12]. Liu Zheng'ai points out that history is the most compelling core selling point of Hetuala City. He believes that the commercialization of historical culture can not only enhance tourism appeal but also achieve a dual improvement in cultural value and economic benefits [13].

In summary, Hetuala City, as a significant historical site marking the early rise of the Qing Dynasty and a key origin of Manchu culture, possesses a comprehensive and distinctive cultural resource system, serving not only as an important component of the national historical and cultural heritage but also as a strategic fulcrum for cultural inheritance and innovation, as well as the promotion of local cultural-tourism integration in the context of the new era.

## 2.2 Tourism cultural and creative products

Tourism cultural and creative products have emerged as a rapidly developing field in recent years, driven by the deep integration of culture and tourism [14]. They refer to innovative products developed on the basis of a tourist destination's unique cultural resources, in which the core cultural characteristics are thoroughly explored and representative design elements are extracted. By applying creative design concepts and modern technological approaches, these products integrate cultural connotations into the tourism consumption experience, thereby creating items that embody cultural value, commemorative significance, and market competitiveness [15]. As a medium, tourism cultural and creative products effectively promote and disseminate the unique culture of a destination, enhancing tourists' awareness and identification with the local culture, while also facilitating the efficient utilization of local cultural resources and fostering innovation in the

cultural tourism industry [16]. Compared to ordinary goods, tourism cultural and creative products not only possess the basic attributes of general merchandise but also embody fundamental characteristics such as cultural relevance, regional identity, narrativity, creativity, and commemorative value [17]. In recent years, China's Ministry of Culture and Tourism has continuously increased support for the development of cultural and creative tourism resources, making the cultural and creative industry a focal topic. Against the backdrop of cultural-tourism integration, tourism cultural and creative products have performed actively in the consumer market, with sales showing rapid growth particularly in regions with developed tourism industries.

Taking the Palace Museum as an example, it leverages its rich cultural resources to develop the "Palace Museum IP," launching numerous creative products targeted at young consumers, achieving an organic integration of history and modernity, culture and technology, tradition and innovation [18]. In 2024, sales of the Palace Museum's tourism cultural and creative products exceeded 2 billion yuan, surpassing ticket revenue of 1 billion yuan, highlighting the immense commercial potential of traditional cultural IP. The National Museum of China drew inspiration from the Nine-Dragon Nine-Phoenix Crown of Empress Xiaoduan of the Ming Dynasty to launch cloisonné-style phoenix crown refrigerator magnets, which sold over one million units and drove sales of the phoenix crown IP series beyond 100 million yuan. Dunhuang has developed tourism cultural and creative products based on murals, flying apsaras, and other elements, covering office and daily life items. The Dunhuang mural blind boxes, combining cultural significance and entertainment, have become a market sensation.

The United Kingdom introduced the concept of "creative industries" in 1988. The British Museum has leveraged its rich collections to launch tourism cultural and creative products such as the Little Yellow Duck, the Rosetta Stone, and the Lewis Chessmen, achieving both cultural dissemination and economic benefits. Since the 1990s, Japan has implemented the "Cultural Nation" and "Cool Japan" initiatives. The Kumamon IP integrates local characteristics with popular culture, gaining widespread popularity through its friendly image and regional symbols, thereby driving local economic growth and becoming a model for tourism revitalization.

In summary, tourism cultural and creative products, as a significant outcome of the deep integration of culture and tourism, are increasingly demonstrating their unique value and broad development potential. From the cultural and creative practices of iconic Chinese cultural landmarks such as the Palace Museum, the National Museum of China, and Dunhuang,to international successful cases such as the British Museum and Kumamon,all demonstrate that tourism cultural and creative products have become important means to promote cultural dissemination, boost tourism consumption, and drive regional economic development. Their design not only embodies cultural connotations and strengthens regional identity,but also achieves the market transformation of cultural value and emotional connection through creative adaptation and modern technological methods, enhancing tourists' cultural experience and sense of identification.

## 2.3  KANO model

The KANO model, proposed by Japanese scholar Noriaki Kano, is a theoretical tool based on analyzing the impact of user needs on satisfaction, designed to reveal the relationship between product performance and user satisfaction [19]. In the study of user needs and design methodologies, the KANO model effectively identifies and categorizes different types of requirements, assisting designers in improving product quality, optimizing user experience, and providing strong support for analysis and decision-making [20]. According to the principles of the KANO model, customer purchase needs can be defined across five levels: Must-be (M), Excitement (A), Performance (O), Indifferent (I), and Reverse (R) quality attributes (Fig 2) [21]. Must-be Quality refers to attributes that do not increase satisfaction when present but cause dissatisfaction when absent. Performance Quality denotes attributes that increase user satisfaction when fulfilled and decrease satisfaction when unmet. Excitement Quality refers to attributes that enhance satisfaction when present but have no impact when absent. Indifferent Quality means attributes whose presence or absence does not affect satisfaction. Reverse Quality

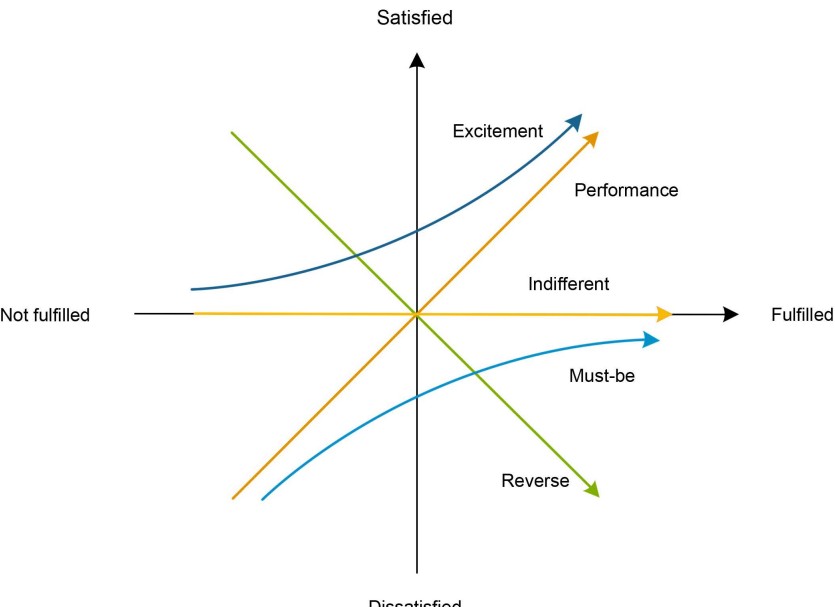

**Fig 2. Relationship between customer satisfaction and requirement fulfillment according to KANO models.**

indicates attributes that decrease satisfaction when present and increase it when absent [22,23]. In recent years, the KANO model has been widely applied in product development and design research. For example, Li Ze, Xie Wang, et al. addressed the developmental constraints of Xiang embroidery products by analyzing consumer demands. Using the KANO model, they classified demand types across four dimensions—visual appearance, functional application, product quality, and intrinsic value—to study consumer preferences for Xiang embroidery tourism cultural and creative products [24]. Zhou You et al. utilized the KANO data model to analyze explicit and latent consumer demands for Jingchu cultural and creative derivatives, identifying key demand points and priorities for the Hubei Provincial Museum's Jingchu cultural products, thereby providing theoretical support and practical guidance for future product design and development [25]. It is evident that the KANO model, with its unique approach to user satisfaction analysis, effectively identifies consumers' multi-level demands in tourism cultural and creative products. Compared to traditional linear satisfaction models, the KANO model emphasizes the asymmetric effects of different types of demands on consumer satisfaction, helping designers uncover users' latent emotional preferences and added value needs, making it especially suitable for the development of tourism cultural and creative products that combine functionality and cultural significance. However, the KANO model also has certain limitations. The model does not explicitly reflect the priority and weight distribution of user needs, which is disadvantageous for subsequent quantitative ranking and resource allocation decisions. Therefore, in practical applications, it is often necessary to complement the KANO model with other methods to enhance the scientific rigor and practicality of demand analysis.

## 2.4 Analytic hierarchy process (AHP)

The Analytic Hierarchy Process (AHP) is an evaluation method that combines qualitative and quantitative analysis using mathematical statistics to address complex decision-making problems [26]. It decomposes decision objectives into multiple hierarchical levels, comprehensively evaluates factors at each level through subjective assessments and statistical data, and determines the relative quality and importance of these factors based on their calculated weights. AHP offers three key advantages: systematization—it views the object as an integrated system and follows a stepwise process of

decomposition, comparison, judgment, and synthesis for decision-making; practicality—it combines qualitative and quantitative approaches to solve problems that traditional optimization methods cannot address; and simplicity—its calculation process is straightforward and results are clear, enabling decision-makers to quickly and directly understand and apply the outcomes [27]. In practical applications, Gao Yuchen et al. collected passenger requirements for aircraft cabin interiors through interviews and questionnaires, used AHP to determine the weights of each demand, and subsequently combined it with Quality Function Deployment (QFD) to effectively translate user needs into cabin design attributes, forming clear design guidelines [28]. Karasan et al. integrated AHP with DEMATEL and QFD models for automotive seat design, employing fuzzy multi-criteria analysis to establish logical relationships between user needs and product functions [29]. Qian Tang et al. applied the KANO model to classify and refine user needs in agricultural intelligent robot design, built an evaluation index system, then introduced a combined weighting approach using AHP and entropy weight method to prioritize demand indicators, ultimately guiding product function optimization and form design [30]. Therefore, AHP demonstrates significant advantages in quantitative analysis for identifying and prioritizing user needs; by constructing pairwise comparison matrices, it effectively enhances the scientific rigor and stability of demand ranking.

The KANO model identifies multi-level needs such as must-be, performance, and excitement qualities from the perspective of user satisfaction, effectively reflecting users' sensitivity and differentiation in perception. However, its judgment of the relative importance weights among different attributes tends to be subjective and lacks a systematic quantitative basis. In contrast, the Analytic Hierarchy Process (AHP) constructs a hierarchical model and incorporates expert judgment matrices, enabling a scientific allocation of weights for various demand items while enhancing the logical consistency and stability of decision outcomes through consistency testing. Therefore, the combined application of the KANO model and AHP not only preserves the KANO model's advantage in sensitively identifying user emotional responses but also leverages AHP to assign scientifically justified weights to user demand items, thereby constructing a more rigorous, systematic, and quantifiable user needs analysis model. This model provides a more reliable decision-making foundation and methodological support for the development of tourism cultural and creative products in Hetuala City.

## 3 Methods

This study collected user demand information through interviews and employed the KANO model to screen and classify the obtained demands, extracting core needs and clarifying consumer demand attributes. Based on this, a KANO demand indicator evaluation system for tourism cultural and creative products was established. On this basis, the Analytic Hierarchy Process (AHP) was introduced. By constructing judgment matrices, weights were assigned to various demand attributes to quantify their importance, thereby identifying critical needs with significant consumer impact and key design directions requiring optimization. This provides scientific grounds and practical guidance for the design and development of tourism cultural and creative products. This study received approval from the Ethics Committee of the School of Art and Design at Liaoning Petrochemical University, with the survey conducted from February 1 to February 15, 2024. All participants signed informed consent forms and voluntarily participated in the study. No minors were involved in the research process. All data and information from participants were anonymized to ensure that no personal identification information was disclosed.

### 3.1 User demand acquisition through interviews

This study collected user demands through interviews. The research team conducted face-to-face conversations with respondents based on a pre-established interview guide, engaging in in-depth discussions related to tourism cultural and creative products. During the interviews, researchers meticulously recorded the information provided by respondents, focusing on capturing key terms mentioned to ensure the comprehensiveness and accuracy of the user demand data.

The data for this study were sourced from visitors to Hetuala City, aiming to capture user demand information specifically related to Hetuala City's tourism cultural and creative products. The study developed a design demand checklist for

tourism cultural and creative products based on a literature review, while considering factors such as the sales scope and consumer demographics of Hetuala City's tourism cultural and creative products to define the survey scope and direction. The survey targeted locations primarily within the Hetuala City tourism area and its surrounding commercial districts. During the survey, a random sampling method was employed to conduct preliminary interviews with tourists and consumers, from which 25 respondents were selected for in-depth interviews to gain a deeper understanding of user needs and preferences. Additionally, the study incorporated expert interviews by convening a panel of 13 specialists in relevant fields to further screen and categorize the collected sample data.

Based on this, 150 tourists were selected as the questionnaire survey sample to obtain broader user feedback. To ensure the validity and representativeness of the survey results, respondents were required to meet the following criteria: aged between 18 and 60, possess a certain level of understanding of tourism cultural and creative products, and have relevant purchasing and usage experience. Additionally, the sample covered various occupational backgrounds and cultural levels to ensure respondents had a certain interest in and willingness to engage with tourism cultural and creative products.

### 3.2 User demand classification based on the KANO model

**3.2.1 Questionnaire design for the KANO model.** Based on the fundamental principles of the KANO model, a user demand KANO questionnaire was developed, in which each user demand is assessed bidirectionally by posing questions regarding the presence or absence of the demand. Respondents rate their feelings on a five-point scale: "Like," "Must be," "Neutral," "Can tolerate," and "Dislike," corresponding to scores of 5, 4, 3, 2, and 1, respectively [31]. The core of this approach lies in separately inquiring about users' feelings when the demand is fulfilled or unfulfilled, enabling an authentic reflection of users' actual perceptions toward the product attributes [32]. Taking "commemorative significance" as an example, the KANO questionnaire format is shown in Table 1.

**3.2.2 KANO model demand analysis.** Based on the results of the KANO questionnaire and in reference to the KANO model evaluation criteria in Table 2, the demand attribute of a specific design requirement indicator for the surveyed participants is determined.

### 3.3 AHP weight calculation

**3.3.1 Construct the hierarchical structure model.** Based on the characteristics of the KANO model and the preliminary analysis results derived from it, combined with the fundamental principles of the AHP (Analytic Hierarchy

**Table 1. KANO survey questionnaire format.**

| KANO Survey Questionnaire | | | | | | | |
|---|---|---|---|---|---|---|---|
| NO. | User Demand | | Like | Definitely | Neither | Mostly unacceptable | Dislike |
| 1 | commemorative significance | Present | 5 | 4 | 3 | 2 | 1 |
| | | Absent | 5 | 4 | 3 | 2 | 1 |

**Table 2. KANO comparison table of model evaluation results classification.**

| Demand Properties | | Answer to functional question | | | | |
|---|---|---|---|---|---|---|
| | | Dislike | Mostly unacceptable | Neither | Definitely | Like |
| Answer to dysfunctional question | Dislike | Q | R | R | R | R |
| | Mostly Unacceptable | M | I | I | I | R |
| | Neither | M | I | I | I | R |
| | Definitely | M | I | I | I | R |
| | Like | O | A | A | A | Q |

Process), a hierarchical analysis structure of tourism cultural and creative product demands was established for expert interviews, as shown in Fig 3.

**3.3.2 Construct the judgment matrix for user requirements.** To calculate the relative importance of each criterion at a given level compared to the previous level, a pairwise comparison method is primarily used to derive the weight values of criteria within the hierarchy, thereby constructing a scientifically sound judgment matrix B [33]. In constructing the judgment matrix, Bij represents the importance rating of each criterion B1, B2, …, Bn relative to the goal B; conversely, 1/Bij is used when the inverse comparison applies. In this judgment matrix, Bij is calculated using the nine-point scale method, employing the numbers 1–9 and their reciprocals to represent scale values, thus enabling quantitative assessment of the relative importance of each criterion [34]. The judgment matrix is shown in Table 3. The meanings of the 1–9 scale levels are shown in Table 4.

**3.3.3 Weight calculation.** Set the judgment matrix be $B = (b_{ij})n*n$. The characteristic vector of the matrix is calculated using the sum-product method, with the following procedure:

(1) Normalize the elements in matrix B by columns: $\bar{b}_{ij} = b_{ij} / \sum_{k=1}^{n} b_{kj}$, i, j = 1, 2, ..., n。

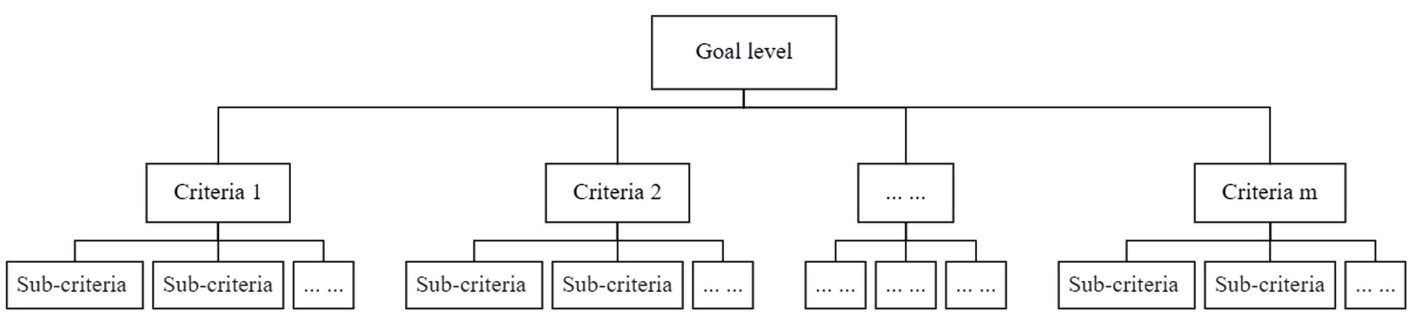

**Fig 3. AHP hierarchical structure format.**

**Table 3. Judgment matrix format.**

| B | $B_1$ | $B_2$ | $B_3$ | ... | $B_n$ |
|---|---|---|---|---|---|
| $B_1$ | $B_{11}$ | $B_{12}$ | $B_{13}$ | ... | $B_{1n}$ |
| $B_2$ | $B_{21}$ | $B_{22}$ | $B_{23}$ | ... | $B_{2n}$ |
| $B_3$ | $B_{31}$ | $B_{32}$ | $B_{33}$ | ... | $B_{3n}$ |
| ... | ... | ... | ... | ... | |
| $B_n$ | $B_{n1}$ | $B_{n2}$ | $B_{n3}$ | ... | $B_{nn}$ |

**Table 4. Questionnaire 1-9 description of the meaning of indicators.**

| Intensity of Importance | Definitions | Explanation |
|---|---|---|
| 1 | Equally important | Two factors have the same importance |
| 3 | Moderately important | One factor is slightly more important than the other |
| 5 | Strongly important | One factor is obviously more important than the other |
| 7 | Very Strongly important | One factor is strongly more important than the other |
| 9 | Extremely Strong important | One factor is extremely more important than the other |
| 2、4、6、8 | Intermediate value between previous levels | Intermediate values of above adjacent comparisons |

(2) Sum the elements of each row across all columns in the normalized matrix: $\widetilde{w}_i = \sum_{j=1}^{n} \bar{b}_{ij}$, i = 1, 2, ..., n。

(3) Divide $\widetilde{w}_i$ by $n$ to obtain the weight vector: $w_i = \widetilde{w}_i / n$。

(4) Solve for the maximum eigenvalue of the judgment matrix: $\lambda_{max} = \frac{1}{n} \sum_{i=1}^{n} \frac{(Bw)_i}{w_i}$。

(5) Conduct a consistency test: $CI = \frac{\lambda_{max} - n}{n-1}$ $CR = \frac{CI}{RI}$。

RI is the Random Consistency Index, and CR represents the Consistency Ratio of the judgment matrix. When CR ≤ 0.1, the judgment matrix is considered reasonably consistent, and the consistency test is passed. If CR > 0.1, the consistency test fails, and the judgment matrix needs to be revised [35].

After passing the consistency test, the comprehensive weights of user requirements are ranked by calculating the product of the weights at the criteria level and the weights at the indicator level, thereby clarifying the relative importance of each requirement indicator in influencing user satisfaction [36].

## 4 Results and discussion

### 4.1 User demand indicators

Using a random sampling method, interviews were conducted with tourists and consumers, from which 25 respondents were selected for in-depth interviews. This study aims to design and develop tourism cultural and creative products and to promote their consumption; therefore, the participants for the in-depth interviews were selected from relevant stakeholders. Among these respondents, three were staff members of Hetuala City, twelve were consumers of tourism cultural and creative products, and ten were designers specializing in product design. During the brainstorming session, consumer demand served as the core theme to facilitate associative thinking, extensively gathering demand-related information concerning tourism cultural and creative products design, and extracting key demand attributes. Based on the market random survey and the in-depth interviews with the 25 respondents, a total of 80 consumer demand samples were collected. However, despite the large quantity and broad diversity of these demand samples, they also exhibited a certain degree of subjectivity and redundancy. To improve data validity and enhance the scientific rigor of subsequent analysis, the study employed the expert interview method to further screen and categorize the sample data. The expert team consisted of three product designers, five university design faculty members, and five product developers from cultural and creative enterprises. The team consolidated synonyms, removed duplicate items, and eliminated invalid information from the collected demand data, ultimately extracting 29 representative core consumer influence factors. Through further classification and integration, these factors were categorized into three primary demand indicators: spiritual needs, material needs, and behavioral needs. The classification results of user demand indicators are presented in Table 5.

### 4.2 Classification of user requirements

Based on the 29 user requirement indicators, a KANO survey questionnaire was developed. The questionnaire survey was distributed online, with a total of 150 copies issued and 150 copies collected. Among the collected questionnaires,

**Table 5. Consumption influencing factors.**

| Spiritual needs | Material level needs | Behavioral needs |
|---|---|---|
| period features, artistic characteristics, commemorative significance, storytelling, national characteristics, history and culture, cultural integration, evoking memories, regional characteristics | series design, modern styling, color design, material design, pattern design, packaging design, aesthetic function, process design, custom design, multi-function, green environment, technology | portable design, interesting features, practical function, easy to mail, price suitable, decorative function, after-sales service, operability |

2 were deemed invalid due to irregular completion or missing information, and 3 contained partial issues but retained certain reference value. Ultimately, 145 questionnaires were determined to be valid, resulting in an effective recovery rate of 96.7%.In terms of demographic characteristics, 56% of respondents were male, and 44% were female. Regarding age distribution, 15.14% were aged 18–26, 35.24% were aged 27–38, 34.07% were aged 39–50, and 15.55% were above 50 years old. The statistical results of the questionnaire are presented in Table 6.

According to the results of the KANO model, there are five Must-be quality indicators (M): history and culture, regional characteristics, modern design, pattern design, and practical functionality. These factors represent the distinctive characteristics of rural culture and constitute the basic and essential requirements for cultural product design. There are nine Performance quality indicators (O): historical period characteristics, commemorative significance, storytelling, national characteristics, material design, packaging design, multi-functionality, portability, and after-sales service. When these needs are fulfilled, consumer satisfaction improves accordingly; conversely, if these needs are not met, visitor satisfaction decreases. There are ten Attractive quality indicators (A): artistic characteristics, cultural integration, evoking memories,

**Table 6. KANO model evaluation results.**

| Classification | No. | Demand Indicators | Quality Dimension | | | | | Quality Type |
|---|---|---|---|---|---|---|---|---|
| | | | A | O | M | I | R | |
| Spiritual needs | 1 | Period features | 10.10% | 38.38% | 3.03% | 31.31% | 17.17% | O |
| | 2 | Artistic characteristics | 66.67% | 21.21% | 3.03% | 4.04% | 2.02% | A |
| | 3 | Commemorative significance | 34.34% | 54.55% | 5.05% | 4.04% | 2.02% | O |
| | 4 | Storytelling | 25.25% | 48.48% | 17.17% | 4.04% | 3.03% | O |
| | 5 | National characteristics | 16.16% | 38.38% | 22.22% | 7.07% | 3.03% | O |
| | 6 | History and culture | 2.02% | 1.01% | 57.58% | 28.28% | 4.04% | M |
| | 7 | Cultural integration | 71.72% | 3.03% | 0.00% | 16.16% | 5.05% | A |
| | 8 | Evoking memories | 81.82% | 2.02% | 1.01% | 8.08% | 4.04% | A |
| | 9 | Regional characteristics | 2.02% | 3.03% | 48.48% | 34.34% | 7.07% | M |
| Material level needs | 10 | Series design | 23.23% | 5.05% | 8.08% | 27.27% | 29.29% | R |
| | 11 | Modern styling | 2.02% | 1.01% | 57.58% | 28.28% | 4.04% | M |
| | 12 | Color design | 73.74% | 6.06% | 0.00% | 16.16% | 4.04% | A |
| | 13 | Material design | 19.19% | 55.56% | 11.11% | 4.04% | 2.02% | O |
| | 14 | Pattern design | 1.01% | 1.01% | 50.51% | 36.36% | 10.10% | M |
| | 15 | Packaging design | 16.16% | 62.63% | 15.15% | 5.05% | 1.01% | O |
| | 16 | Aesthetic function | 80.81% | 5.05% | 1.01% | 6.06% | 6.06% | A |
| | 17 | Process design | 72.73% | 1.01% | 0.00% | 11.11% | 4.04% | A |
| | 18 | Custom design | 4.04% | 2.02% | 20.20% | 27.27% | 43.43% | R |
| | 19 | Multi-function | 14.14% | 63.64% | 10.10% | 2.02% | 5.05% | O |
| | 20 | Green Environment | 71.72% | 1.01% | 0.00% | 10.10% | 7.07% | A |
| | 21 | Technology | 27.27% | 5.05% | 15.15% | 26.26% | 23.23% | A |
| Behavioral needs | 22 | Portable design | 16.16% | 46.46% | 13.13% | 6.06% | 1.01% | O |
| | 23 | Interesting features | 72.73% | 3.03% | 1.01% | 6.06% | 9.09% | A |
| | 24 | Practical function | 0.00% | 1.01% | 63.64% | 25.25% | 6.06% | M |
| | 25 | Easy to mail | 64.65% | 1.01% | 1.01% | 26.26% | 4.04% | A |
| | 26 | Price suitable | 20.20% | 6.06% | 13.13% | 26.26% | 21.21% | I |
| | 27 | Decorative function | 8.08% | 2.02% | 22.22% | 36.36% | 29.29% | I |
| | 28 | After-sales service | 14.14% | 54.55% | 16.16% | 1.01% | 3.03% | O |
| | 29 | Operability | 20.20% | 6.06% | 7.07% | 30.30% | 29.29% | I |

color design, aesthetic functionality, craftsmanship design, environmental friendliness, technological features, playful features, and ease of mailing. Attractive needs reflect the innovative aspects of cultural product design and often involve new methods and technologies. When attractive needs are met, tourist satisfaction increases significantly; however, their absence does not lead to a decline in satisfaction. Indifferent quality indicators (I) are not considered, as tourists hold a neutral attitude toward these needs.

## 4.3  AHP calculation of user demand weights

Based on the classification results of the KANO model, the study further integrates the AHP method to construct a judgment matrix, conduct pairwise comparisons of the importance among user demand indicators within the same level, and calculate weights using the geometric mean method, ultimately deriving the relative weights of the criteria layer and the indicator layer. In this integrated model, a top-down, hierarchical decomposition approach is used to build a three-level user demand evaluation indicator system. The first level is the goal layer, namely "Design of Hetuala City Tourism Cultural and Creative Products";The second level is the criteria layer, corresponding to the three core demand types identified in the KANO model—Must-be (M), Performance (O), and Attractive (A) qualities;The third level is the indicator layer, where under the M, O, and A demand categories, a total of 24 specific user demand indicators are further refined. The hierarchical structure is shown in Fig 4.

Based on the Analytic Hierarchy Process (AHP) model, a scientifically rational judgment matrix was constructed. The judgment matrix enables pairwise comparisons of the relative importance among elements within the same level. Subsequently, multiple experts were invited to assess the relative importance of user demands at the same level by assigning

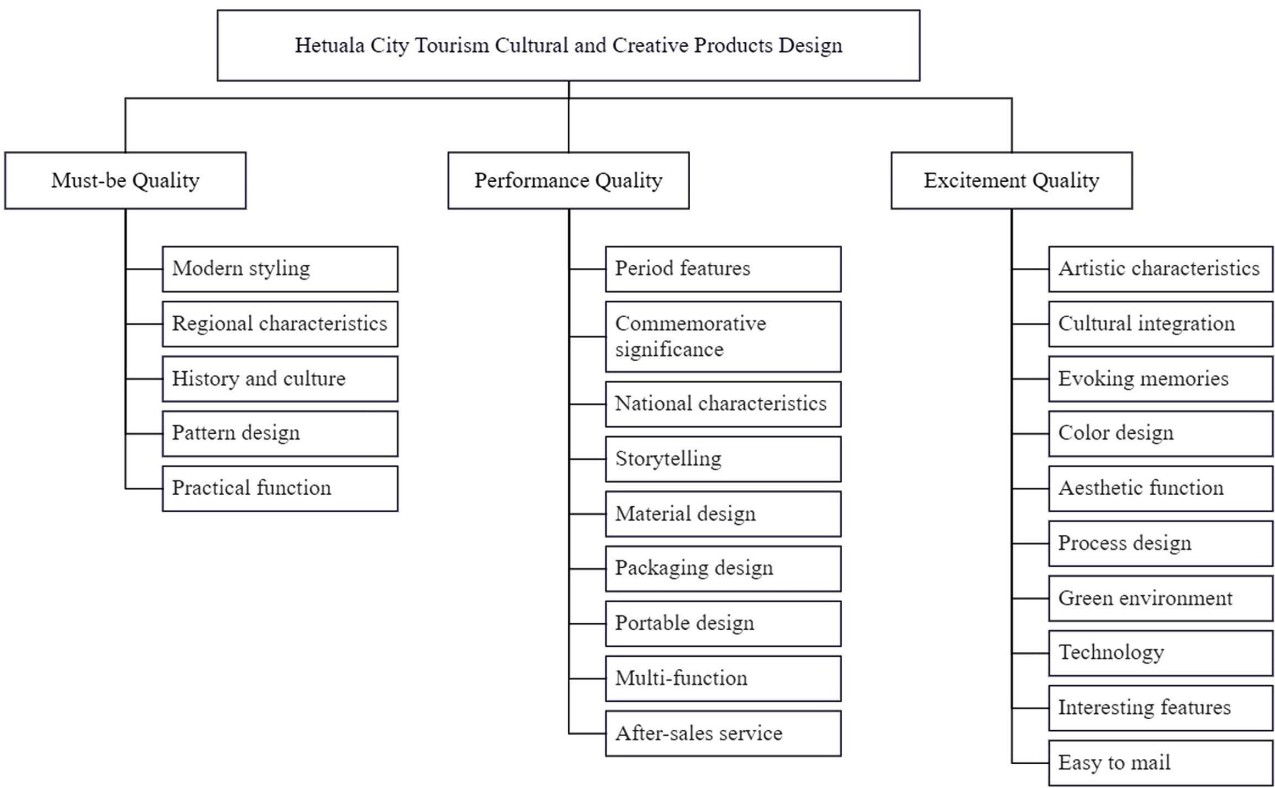

**Fig 4.  Hierarchical structure of demand indicators.**

values according to a 1-to-9 scale [37]. To ensure the validity of the evaluation results, an expert panel of 10 members with extensive practical and research experience in the field of tourism cultural creative products was convened, including 5 experts in tourism management and 5 product designers. Based on the experts' scores, the average values were calculated. Using the judgment matrix in combination with the geometric mean method, the judgment matrices and corresponding weights for each level were computed. The results are presented in Tables 7–10.

To evaluate the stability and reliability of the judgment matrix, ensuring that the weights or preferences do not exhibit significant contradictions, a consistency test must be conducted. The test is considered passed when CR ≤ 0.1. According to the results calculated by SPSS software, the CR values of all raters are below 0.1, indicating that the consistency test has been passed. The results of the consistency test are shown in Table 11.

After completing the consistency test, the comprehensive weights of user requirements were ranked by calculating the product of the weights of criteria-level indicators and those of sub-criteria-level indicators, thereby clarifying the importance of each requirement indicator's impact on user satisfaction. The results of the comprehensive weight calculations are presented in Table 12.

Based on the results of Table 6 (KANO user demand classification) and Table 11 (AHP comprehensive weight importance ranking), it can be observed that regional characteristics rank first in the AHP comprehensive weight order and are classified as must-be quality attributes in the KANO model. This indicates that this demand is not only highly important

**Table 7. Primary demand indicator judgment matrix.**

| Tier 1 Demand Indicators | Must-be Quality | Performance Quality | Excitement Quality |
|---|---|---|---|
| Must-be Quality | 1 | 5 | 7 |
| Performance Quality | 1/5 | 1 | 3 |
| Excitement Quality | 1/7 | 1/3 | 1 |

**Table 8. Must-be demand indicator judgment matrix.**

| Must-be Quality | Modern styling | Regional characteristics | History and culture | Pattern design | Practical function |
|---|---|---|---|---|---|
| Modern styling | 1 | 1/9 | 1/7 | 1/3 | 1/5 |
| Regional characteristics | 9 | 1 | 3 | 6 | 4 |
| History and culture | 7 | 1/3 | 1 | 5 | 3 |
| Pattern design | 3 | 1/6 | 1/5 | 1 | 1/3 |
| Practical function | 5 | 1/4 | 1/3 | 3 | 1 |

**Table 9. Excitement demand index judgment matrix.**

| Excitement Quality | Artistic characteristics | Cultural integration | Evoking memories | Color design | Aesthetic function | Process design | Green environment | Technology | Interesting features | Easy to mail |
|---|---|---|---|---|---|---|---|---|---|---|
| Artistic characteristics | 1 | 1/6 | 2 | 3 | 1/2 | 3 | 3 | 1/2 | 1/3 | 2 |
| Cultural integration | 6 | 1 | 4 | 3 | 3 | 4 | 4 | 2 | 2 | 5 |
| Evoking memories | 1/2 | 1/4 | 1 | 1/2 | 1/3 | 1/2 | 1/2 | 1/3 | 1/4 | 1/2 |
| Color design | 1/3 | 1/3 | 2 | 1 | 1/2 | 2 | 2 | 1/3 | 1/4 | 4 |
| Aesthetic function | 2 | 1/3 | 3 | 2 | 1 | 3 | 3 | 1/2 | 1/3 | 4 |
| Process design | 1/3 | 1/4 | 2 | 1/2 | 1/3 | 1 | 3 | 1/2 | 1/3 | 4 |
| Green Environment | 1/3 | 1/4 | 2 | 1/2 | 1/3 | 1/3 | 1 | 1/2 | 1/3 | 2 |
| Technology | 2 | 1/2 | 3 | 3 | 2 | 2 | 2 | 1 | 1/2 | 3 |
| Interesting features | 3 | 1/2 | 4 | 4 | 3 | 3 | 3 | 2 | 1 | 4 |
| Easy to mail | 1/2 | 1/5 | 2 | 1/4 | 1/4 | 1/4 | 1/2 | 1/3 | 1/4 | 1 |

**Table 10. Performance demand index judgment matrix.**

| Performance Quality | Period features | Commemorative significance | National characteristics | Storytelling | Material design | Packaging design | Portable design | Multi-function | After-sales service |
|---|---|---|---|---|---|---|---|---|---|
| Period features | 1 | 1/4 | 1/4 | 1/6 | 4 | 3 | 1/2 | 1/5 | 2 |
| Commemorative significance | 4 | 1 | 2 | 1/4 | 4 | 3 | 1/2 | 1/3 | 3 |
| National characteristics | 4 | 1/2 | 1 | 1/4 | 4 | 3 | 2 | 1/2 | 3 |
| Storytelling | 6 | 4 | 4 | 1 | 5 | 4 | 2 | 2 | 3 |
| Material design | 1/4 | 1/4 | 1/4 | 1/5 | 1 | 5 | 1/3 | 1/4 | 3 |
| Packaging design | 1/3 | 1/3 | 1/3 | 1/4 | 1/5 | 1 | 1/3 | 1/4 | 3 |
| Portable design | 2 | 2 | 1/2 | 1/2 | 3 | 3 | 1 | 1/3 | 3 |
| Multi-function | 5 | 3 | 2 | 1/2 | 4 | 4 | 3 | 1 | 4 |
| After-sales service | 1/2 | 1/3 | 1/3 | 1/3 | 1/3 | 1/3 | 1/3 | 1/4 | 1 |

**Table 11. Consistency test results.**

| Summary of Consistency Test Results | | | | | |
|---|---|---|---|---|---|
| | λmax | CI value | RI value | CR value | Consistency test results |
| Primary indicator requirements | 3.065 | 0.032 | 0.525 | 0.062 | Qualified |
| Must-be Quality (M) | 5.23 | 0.058 | 1.11 | 0.052 | Qualified |
| Excitement Quality (A) | 10.88 | 0.098 | 1.486 | 0.066 | Qualified |
| Performance Quality (O) | 10.129 | 0.141 | 1.451 | 0.097 | Qualified |

**Table 12. Comprehensive weight calculation results.**

| Primary demand indicators | Weights | Secondary demand indicators | Weighting value | Combined weights | Order |
|---|---|---|---|---|---|
| Must-be Quality | 0.72351 | Regional characteristics | 0.47978 | 0.34713 | 1 |
| | | History and culture | 0.27012 | 0.19543 | 2 |
| | | Practical function | 0.14369 | 0.10396 | 3 |
| | | Pattern design | 0.07132 | 0.05160 | 4 |
| | | Modern styling | 0.03509 | 0.02539 | 7 |
| Performance Quality | 0.19319 | Storytelling | 0.25936 | 0.05011 | 5 |
| | | Multi-function | 0.20443 | 0.03949 | 6 |
| | | National characteristics | 0.12052 | 0.02328 | 8 |
| | | Commemorative significance | 0.10975 | 0.02120 | 9 |
| | | Portable design | 0.10897 | 0.02105 | 10 |
| | | Period features | 0.06443 | 0.01245 | 13 |
| | | After-sales service | 0.05106 | 0.00986 | 15 |
| | | Material design | 0.04849 | 0.00937 | 16 |
| | | Packaging design | 0.03301 | 0.00638 | 19 |
| Excitement Quality | 0.08331 | Cultural integration | 0.24041 | 0.02003 | 11 |
| | | Interesting features | 0.18299 | 0.01524 | 12 |
| | | Technology | 0.12359 | 0.01030 | 14 |
| | | Aesthetic function | 0.10860 | 0.00905 | 17 |
| | | Artistic characteristics | 0.08836 | 0.00736 | 18 |
| | | Color design | 0.07075 | 0.00589 | 20 |
| | | Process design | 0.06713 | 0.00559 | 21 |
| | | Green Environment | 0.04776 | 0.00398 | 22 |
| | | Easy to mail | 0.03566 | 0.00297 | 23 |
| | | Evoking memories | 0.03474 | 0.00289 | 24 |

to consumers but also an indispensable foundational component in cultural creative product design. The prominence of regional characteristics enhances consumers' perception of local identity and cultural belonging; therefore, it should be emphasized and deeply explored in the design implementation.

Historical and cultural attributes also belong to must-be quality attributes, ranking second in comprehensive weight. This implies that incorporating representative historical and cultural elements in products can not only meet consumers' basic expectations but also possess strong appeal and market recognition. During the design process, historical and cultural information should be materialized through multiple dimensions such as patterns, craftsmanship, and materials to enhance the cultural value density of the product.

Regarding excitement quality attributes, cultural integration ranks 11th in the AHP comprehensive ranking but holds the highest importance within the excitement attributes in the KANO model. This suggests that after meeting basic functions, this element can significantly increase consumer satisfaction. Design efforts should actively explore the fusion and innovation of various cultural forms, including traditional and modern, East and West, ethnic and mainstream cultures, to enhance the product's emotional appeal and aesthetic differentiation.

The comprehensive weight ranking of ethnic characteristics is 8th, indicating a certain level of importance. Combined with the KANO model, ethnic characteristics can be regarded as a key intermediary connecting regional features and cultural integration; thus, ethnic symbols, folk customs, and linguistic elements should be reflected in product details to strengthen cultural recognition.

Storytelling demand is classified as a performance quality attribute, ranking first within this category and fifth in overall comprehensive weight. It is evident that products with a certain level of plot or narrative structure can help evoke emotional resonance and user identification. Therefore, product design can incorporate regional cultural backgrounds to create product storylines and enhance user immersion.

Practicality, as a must-be quality attribute, ranks third overall, indicating that consumers maintain high attention to the functional use of cultural creative products. Compared to additional cultural symbols and artistic expressions, the realization of basic functions is the primary condition for product market acceptance.

Multifunctionality is classified as a performance quality attribute, ranking second within that category and sixth overall, indicating that users expect products to maintain basic functions while expanding usage scenarios and additional features.

Fun features and technological aspects are both classified as excitement quality attributes, ranking second and third within this category, respectively. Although their AHP comprehensive rankings are relatively low (12th and 14th), they still reflect users' potential expectations regarding product interactivity and intelligence. Future designs may consider integrating AR/VR, smart modules, and other technologies to enhance user engagement and technological experience.

Other demand factors generally scored low in the AHP comprehensive weight ranking and showed low user sensitivity in the KANO model, indicating that these factors currently have no significant influence on user cognition and purchasing decisions.

## 5 Conclusion and recommendations

With the vigorous development of China's "Culture + Tourism" industry, tourism cultural and creative products have become important carriers for attracting tourists and enhancing destination appeal. However, Hetuala City's cultural and creative products still face certain limitations in content expression and functional design, making it difficult to fully meet the diverse consumption demands of tourists. This study, based on a user-centered design philosophy, constructs a development evaluation framework for tourism cultural and creative products by integrating the KANO model and the Analytic Hierarchy Process (AHP). The results indicate that the KANO model effectively identified three types of demand attributes among Hetuala City tourists: must-be, performance, and excitement qualities. The AHP method was applied to evaluate and rank the weights of demand indicators, clarifying the core design factors and their relative importance, including regional characteristics, historical value, multiculturalism, storytelling, practicality, playfulness, and technological

integration. The construction and application of the KANO-AHP hybrid model not only enhanced the user orientation and scientific rigor of the design process but also provided a practical theoretical tool and implementation pathway for the innovative transformation of local cultural resources.

Based on the above research conclusions, the design and development of Hetuala City's tourism cultural and creative products can be approached from the following seven aspects:

### (1) Emphasize Regional Characteristics and Strengthen Cultural Identity

Regional characteristics constitute the core competitiveness of tourism cultural and creative products and serve as key differentiators from ordinary goods. The design of Hetuala City's tourism cultural and creative products should be rooted in local regional culture, extracting design elements from architectural styles, environmental features, and folk customs. Regional culture, folk traditions, and scenic resources should be transformed into design languages such as shapes, patterns, motifs, colors, and materials, thereby constructing a cultural symbol system with strong regional identification. By deeply exploring Hetuala's regional cultural genes, tourism products that integrate regionality, cultural significance, creativity, and uniqueness can be developed.

### (2) Incorporate Historical Elements to Enhance Cultural Value

As the first capital of the Later Jin dynasty, Hetuala City possesses rich Manchu historical and cultural resources. Product design should be centered on the unique historical culture of the ancient city, transforming historical elements such as architecture, decoration, and folk rituals into modern product forms. Techniques like scene restoration, proportional adjustment, and functional re-creation should be employed to enable consumers to experience a historical atmosphere during use, thereby deepening the cultural experience and enhancing the sense of value.

### (3) Strengthening Multicultural Integration

Hetuala City is located in a multi-ethnic region, where the Manchu, Korean, Hui, Mongolian, and other ethnic groups collectively form a diverse cultural landscape. Product development should transcend the limitations of a single ethnic culture by integrating the distinctive features of Northern ethnic cultures with those of the Central Plains, reflecting this fusion in form, materials, and functionality. This approach fosters a multicultural expression, enhancing both the inclusivity and appeal of the products.

### (4) Enhancing Narrative and Emotional Expression

Cultural creative products are not merely physical carriers but also transmitters of emotions and memories. It is recommended to embed narrative elements in the design, which may originate from the historical context of the artifacts themselves or from the legends associated with their transmission. For products where the narrative is not immediately apparent, textual, graphical, or other storytelling elements can be incorporated into the packaging or product details. Such emotionally-oriented design enhances consumer resonance and purchase intention.

### (5) Emphasizing Practical Functionality and Diversified Design

While fulfilling cultural expression, products should possess strong practicality and multifunctionality. During the development process, it is essential to consider usage scenarios, user habits, and aesthetic preferences to ensure that the product meets daily functional needs while also conveying cultural symbolism. This approach encourages long-term consumer use and facilitates ongoing cultural dissemination.

### (6) Incorporating Interactive and Engaging Experiences

The current market offers an insufficient supply of interactive cultural creative products. It is recommended to enhance product design with greater entertainment value and interactivity, enabling consumers to engage in cultural dissemination

through the processes of use, experience, and sharing. By incorporating innovative forms, dynamic functional interactions, and creative play methods, such designs can cater to the younger generation's pursuit of personalization, fashion, and novelty.

**(7) Integrating Modern Technology to Enhance Innovation**

New technologies, techniques, and materials should be fully utilized to integrate ancient culture with modern technology, endowing products with new functions and experiential qualities. The incorporation of technological elements can not only enhance the innovative value and practical convenience of the products,but also create a striking visual impact that blends the ancient and the modern, thereby enhancing the uniqueness and market competitiveness of Hetuala City's cultural creative products.

Although this study developed an evaluation framework for tourism cultural and creative product development that integrates the KANO model and the AHP method, effectively identifying and quantifying the multi-level demand characteristics of tourists to Hetula City and providing both theoretical foundations and practical pathways for the precise design of tourism cultural and creative products, several limitations remain in the research process, which can be addressed in future studies.

**(1) Limitation of the Research Area**

This study takes Hetuala City as a case, with survey respondents primarily consisting of local tourists, whose cultural backgrounds, travel motivations, and consumption preferences exhibit certain regional specificities. Therefore, the applicability of the research findings may vary when applied to other regions or tourism destinations of different cultural types. Future research should expand the sample scope and conduct cross-regional and cross-cultural comparative analyses to verify the generalizability and adaptability of the model.

**(2) Simplification of the Model Structure**

To enhance practicality and operability, this study adopts a simplified design for the integrated KANO–AHP model, without incorporating additional dynamic variables or contextual factors such as tourist behavior trajectories, consumption conversion rates, or emotional feedback, which may limit the model's ability to reflect real-world conditions. Future research could integrate multi-source analytical methods such as big data mining, machine learning, and fuzzy comprehensive evaluation to improve the model's accuracy and level of intelligence.

**(3) Lack of Design Practice and Empirical Validation**

This study focuses on theoretical model construction and methodological application, without integrating specific tourism cultural and creative product development cases for practical validation, thus lacking an assessment of the model's effectiveness in real-world design processes. Future work should extend the model into practical stages, conducting product development and user testing based on analytical results, and establishing a closed-loop mechanism of "theory–design–validation–optimization" to enhance the model's practical guidance value.

## Author contributions

**Investigation:** Zhaolong Liu.

**Methodology:** Honghe Gao.

**Writing – original draft:** Zhaolong Liu.

**Writing – review & editing:** Zhaolong Liu, Honghe Gao.

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
