## [Decision Letter · Decision Letter 0]

26 Jun 2025

Dear Dr. zhaolong,

We look forward to receiving your revised manuscript.

Kind regards,

Kizito Ogedi Alakwe, Ph.D

Guest Editor

PLOS ONE

“This work was supported by Educational Commission of Liaoning Province of China (No. W2019013)�Liaoning Provincial Education Science "13th Five-Year Plan" Period (No. JG20DB276).”

“This work was supported by Educational Commission of Liaoning Province of China (No. W2019013)�Liaoning Provincial Education Science "13th Five-Year Plan" Period (No. JG20DB276).”

“This work was supported by Educational Commission of Liaoning Province of China (No. W2019013)�Liaoning Provincial Education Science "13th Five-Year Plan" Period (No. JG20DB276).”

Additional Editor Comments:

Thank you for your submission and for the thoughtful engagement with the topic. Having reviewed the manuscript alongside the peer reviewers’ comments, I would like to highlight a few areas that require your attention:

1. Conceptual Clarity: It is imperative that the key concepts used in the paper are clearly defined and situated within the relevant literature. This will enhance the conceptual grounding of your arguments.

2. Methodological Rigor: A dedicated methodology section should be included. Please ensure that your research design, data collection, and analytical approach are clearly articulated. This will significantly strengthen the paper’s academic robustness.

3. Presentation of Findings: As noted by one of the reviewers, the findings should be presented in a structured manner. Consider reworking this section for clarity, highlighting key insights and linking them more explicitly to your research questions or objectives.

4. Referencing Style: Kindly ensure that all citations and references conform to a uniform and accepted referencing style throughout the manuscript.

Please incorporate these points along with the detailed reviewer suggestions in your revision.

Reviewers' comments:

Reviewer's Responses to Questions

**Comments to the Author**

1. Is the manuscript technically sound, and do the data support the conclusions?

Reviewer #1: Partly

Reviewer #2: Partly

2. Has the statistical analysis been performed appropriately and rigorously?

Reviewer #1: I Don't Know

Reviewer #2: Yes

3. Have the authors made all data underlying the findings in their manuscript fully available?

Reviewer #1: Yes

Reviewer #2: Yes

4. Is the manuscript presented in an intelligible fashion and written in standard English?

Reviewer #1: No

Reviewer #2: Yes

Reviewer #1: Thanks for submitting your article. It is clear that your research seeks to understand how the KANO-AHP model can be used by cities such as Hetuala in understanding the demand for creative cultural products.

However, there is a need to establish the concept of creative cultural products in your paper (linked to place branding) as this is a major focus of this research - what are these products? what are the categories? why do cities need them? how have other cities used them? what does Hetuala currently have? who are the current designers of such products in Hetuala? - residents or foreigners, who are the customers of these products? what are the social and economic benefits to the residents or the merchandisers of these products? is there another city doing this well that can be used as a case-in-point? Currently, there has been more focus on the models used in justifying the objective of your study but you still need to situate your paper in the field of place branding/ cultural tourism.

Please see below more comments;

Abstract

- Please include the methodology used in this research. The interviews and survey conducted.

- Please, remain consistent with the language ' cultural and creative' or ' creative cultural products' (This applies to the whole work)

Introduction

- Please define what you mean by cultural/creative products and give xamples of what you mean by cultural and creative products. Do you mean souvenirs, keepsakes or food or folktales or clothing or experiences that evoke the culture of the host country or city? what are some of those global examples? It is recommended that this is established in the introduction. What are the social or economic or political benefits of cultural and creative products in literature? why is it important for a city? does it boost the gdp/ tourism/ brand perception etc?

- You have identified the situation with the type of cultural and creative products in Hetuala, but not referenced the source of this information. Was this research conducted by the authors through participant observation or is available in literature- please reference.

- Most of the concepts in this paper need to be better justified with relatable examples. For example, when you say daily necessities in section 1.2, you need to give examples of the food and clothing as is relevant to Hetuala city. As this shows the uniqueness of the cultural and creative practices. Also merely listing the buildings unique to the city doesn't show the cultural significance. Is there something about some of these buildings that stand out? e.g the spiritual significance or the age of the building? Are they linked to cultural mythologies that can be further explored for the branding of the city and affiliated cultural and creative products?

Methodology

- There should be a methodology section that clearly states what method you have used, this has been briefly done in 3.1.1 but doesnt give a clear justification as to why you have selected tourists (current or prospective?) or product designers (are they all based in Hetuala?). In the city branding concept, there are other stakeholders such as govt administrators, residents, media, industry stakeholders (arts and crafts merchandisers etc), so is there any reason why you have chosen only tourists and product designers for thsi study? Somewhere in your introduction should establish that since you are looking at consumption, you will be focused on the stakeholders that produce and are likely to consumer cultural and creative products.

Presentation of findings

- The presentation of findings should focus on the responses rather than explaining the KANO model in detail. The results of the analysis should show the results from the interview and questionnaire but it is not clear how the interviews with the product designers have been used here.

- Key sections of your work need to be referenced e.g the paragraph on the Qing dynasty and the language spoken in Hetuala. This also applies tot he section on storytelling.

References and Language

- Please revisit your work and include references for some of the claims that have been made in this research

- Also revise language and grammar for more clarity

Other recommendations

Pls consult other works on cultural and creative products

- Design of Tourism Cultural and Creative Products Based on

Regional Historical and Cultural Elements. Yinglu Wu1 (2021)

- Place branding and local food souvenirs: the ethical attributes of national parks’ brands

(Lucia Pizzichini, Valerio Temperini and Gian Luca Gregori, 2020 )

- The use of intangible heritage and creative industries as a tourism asset in the UNESCO creative cities network (Jordi Arcos-Pumarola, Alexandra Georgescu Paquin and Marta Hernández Sitges, 2023)

All the best

Reviewer #2: I feel that the writer did the job to the best of his or her ability. However, the structure of the work could have been improved with a more logical sequence in arranging the sections of the writing using the common IMRAD approach: Introduction, Literature Review, Methodology, Results, and Discussion. Additionally, APA 7th edition should be applied consistently in both in-text citation and referencing (assuming that is required). Furthermore, some of the ideas related to the history of the city that are currently placed in the discussion section should be included in the introduction and literature review sections of the text.

**Do you want your identity to be public for this peer review?** For information about this choice, including consent withdrawal, please see our Privacy Policy

Reviewer #1: No

Reviewer #2: **Yes: ** Nzeaka, Emmanuel Ezimako

---

## [Author Response · Author response to Decision Letter 1]

25 Aug 2025

Dear Academic Editor and Reviewers

We sincerely thank you for your thorough review of our manuscript and for your valuable comments. Your suggestions have been extremely helpful in improving the quality of our research and the manuscript. In response to the reviewers’ feedback, we have thoroughly revised the manuscript. The main modifications are summarized as follows:

1.Manuscript Structure Adjustment: The overall structure of the manuscript has been reorganized into five sections: Introduction, Literature Review, Methodology, Results and Discussion, and Conclusion and Recommendations.

1�Literature Review: A Literature Review section has been added, providing conceptual definitions and explanations for “tourism cultural and creative products,” the KANO model, and the AHP method, along with relevant studies by domestic and international scholars. In addition, the differences and connections between KANO and AHP have been clarified to help readers better understand the theoretical foundation of the study and the rationale for the chosen methodology.

2�Methodology: A dedicated Methodology section has been added, providing a detailed description of the research design, implementation process, and analytical steps. The specific application procedures and operational details of the KANO and AHP methods have been clearly explained, enhancing both transparency and methodological rigor.

3�Results and Discussion: The Results section has been reorganized and structured. In the Discussion section, comparisons and interpretations based on KANO and AHP analysis results have been added, ensuring that the research findings directly address the research questions.

4�Conclusion and Recommendations: A Recommendations subsection has been added. Based on the research findings, specific suggestions for the design and development of tourism cultural and creative products have been provided to enhance the practical applicability and relevance of the study.

2.Title and Abstract Revision: The manuscript title and abstract have been revised. The new abstract has been rewritten in concise and clear academic language, highlighting the research objectives, methods, and key conclusions, ensuring that it accurately reflects the overall scope and contribution of the study.

3.Citations and Reference Formatting: All in-text citations and references have been thoroughly checked and standardized according to APA 7th edition guidelines. Some references have been updated or supplemented to better connect the study with existing scholarly work, thereby improving the manuscript’s academic rigor and completeness.

Response to the Academic Editor

1.Funding Statement: The funding statement has been revised and is now provided in detail within the cover letter.

2.Ethics Statement: The ethics statement has been moved to the Methods section of the paper.

3.Conceptual Definitions: A dedicated section in the Literature Review has been added to define the concepts of “tourism cultural and creative products,” the KANO model, and the AHP method. Relevant references have been included to support these definitions, thereby strengthening the theoretical foundation of the study.

4.Methodology: A separate Methodology section has been added, providing detailed descriptions of the research process, steps, and implementation procedures to improve transparency and reproducibility.

5.Results: The Results section has been revised to provide a systematic analysis based on the findings from the KANO and AHP models, ensuring that the discussion is more data-driven and logically consistent.

6.In-text Citations and Reference Format: All in-text citations and the reference list have been thoroughly checked and revised to ensure full compliance with the journal’s formatting requirements.

7.Data availability statement: All relevant data are within the manuscript.

Response to the Reviewer #1

1.Literature Review: We have added a dedicated Literature Review section, which defines the concept of “tourism cultural and creative products” and illustrates, with examples, their role in promoting regional economic development. Additionally, the Introduction section has been updated to summarize the current design and sales status of tourism cultural and creative products in Hetu Ala City, enhancing the clarity and completeness of the research background.

2.Language Consistency: All references to “tourism cultural and creative products” throughout the manuscript have been standardized to ensure consistent language style, improving readability and overall professionalism.

3.Abstract: The Abstract has been revised to provide a detailed description of the research objectives, methods, results, and conclusions, making it more comprehensive and structured, and enabling readers to quickly grasp the core content of the study.

4.Methodology: A dedicated Methodology section has been added, providing detailed descriptions of the research design, implementation process, and analytical steps. The specific application procedures and operational details of the KANO and AHP methods have been clearly explained to enhance the transparency and rigor of the methodology.

5. Research Results: The Results section has been reorganized and structured. The findings from the KANO and AHP analyses are compared and interpreted systematically, making the research outcomes more clearly aligned with the research questions.

6.Interviewee Selection: Interview participants were selected based on stakeholder relevance, including tourists, cultural and creative product sales personnel, consumers, and professionals in the design industry. The tourist sample covers a range of occupations (e.g., villagers, students, teachers, workers), while in-depth interviews focused on more experienced sales personnel, consumers, and design industry professionals to obtain more precise and professional insights into consumer needs.

7.References: Citations have been revised for sections defining concepts and presenting arguments to ensure that all claims are properly supported. The reference format has also been adjusted to comply with the journal’s guidelines.

Response to the Reviewer #2

1.Manuscript Structure Adjustment: We have optimized and revised the overall structure of the manuscript. The revised structure now includes Introduction, Literature Review, Methodology, Results and Discussion, and Conclusion and Recommendations, which improves the clarity of the research logic and enhances the overall organization.

2. In-text Citations and Reference Format: We have thoroughly revised all in-text citations and the reference list to ensure full compliance with the journal’s formatting requirements.

3.Discussion Section Revision: The Discussion section has been reorganized by integrating its original content with the Literature Review. This makes the discussion more systematic and strengthens the connection between our research findings and existing literature, thereby enhancing the rigor and scholarly value of the arguments.

---

## [Editor Report · Decision Letter 1]

23 Sep 2025

Dear Dr. zhaolong,

Thank you for submitting your manuscript to PLOS ONE. After careful consideration, we feel that it has merit but does not fully meet PLOS ONE’s publication criteria as it currently stands. Therefore, we invite you to submit a revised version of the manuscript that addresses the points raised during the review process.

We look forward to receiving your revised manuscript.

Kind regards,

Kizito Ogedi Alakwe, Ph.D

Guest Editor

PLOS ONE

Journal Requirements:

Additional Editor Comments:

Thank you for submitting your manuscript to this journal. The reviewers and I found the paper topical and engaging; however, it requires substantial revisions before it can be considered for publication.

I encourage you to carefully address the reviewers’ comments and suggestions, as these will significantly strengthen the quality, clarity, and overall contribution of your work. Once revised, your manuscript should be resubmitted for further consideration.

We appreciate the effort you have put into this study and look forward to receiving your thoroughly revised version.

---

## [Author Response · Author response to Decision Letter 2]

22 Oct 2025

The reference suggested by the reviewer has been replaced in the relevant citations of this paper. This section primarily serves to cite others’ definitions and explanations of concepts, rather than as a core basis for argumentation; therefore, the suggested reference was not essential. To ensure the accuracy and appropriateness of the content, more suitable references have now been adopted.

---

## [Editor Report · Decision Letter 2]

26 Nov 2025

Design Strategies for Tourism Cultural and Creative Products of Hetuala City Based on KANO and AHP

PONE-D-25-11994R2

Dear Dr. zhaolong,

We’re pleased to inform you that your manuscript has been judged scientifically suitable for publication and will be formally accepted for publication once it meets all outstanding technical requirements.

Kind regards,

Kizito Ogedi Alakwe, Ph.D

Guest Editor

PLOS ONE
---

## [Editor Report · Acceptance letter]

PONE-D-25-11994R2

PLOS ONE

Dear Dr. Liu,

I'm pleased to inform you that your manuscript has been deemed suitable for publication in PLOS ONE. Congratulations! Your manuscript is now being handed over to our production team.

Kind regards,

on behalf of

Dr. Kizito Ogedi Alakwe

Guest Editor

PLOS ONE